# Intracellular Delivery of siRNAs Targeting AKT and ERBB2 Genes Enhances Chemosensitization of Breast Cancer Cells in a Culture and Animal Model

**DOI:** 10.3390/pharmaceutics11090458

**Published:** 2019-09-03

**Authors:** Tahereh Fatemian, Hamid Reza Moghimi, Ezharul Hoque Chowdhury

**Affiliations:** 1Jeffry Cheah School of Medicine and Health Sciences, Monash University Malaysia, Jalan Lagoon Selatan, 46150 Bandar Sunway, Malaysia; 2School of Pharmacy, Shahid Beheshti University of Medical Sciences, 19839-63113 Tehran, Iran

**Keywords:** siRNA, drug delivery, nanoparticle, carbonate apatite, ERBB2, AKT, breast cancer

## Abstract

Pharmacotherapy as the mainstay in the management of breast cancer suffers from various drawbacks, including non-targeted biodistribution, narrow therapeutic and safety windows, and also resistance to treatment. Thus, alleviation of the constraints from the pharmacodynamic and pharmacokinetic profile of classical anti-cancer drugs could lead to improvements in efficacy and patient survival in malignancies. Moreover, modifications in the genetic pathophysiology of cancer via administration of small nucleic acids might pave the way towards higher response rates to chemotherapeutics. Inorganic pH-dependent carbonate apatite (CA) nanoparticles were utilized in this study to efficiently deliver various classes of therapeutics into cancer cells. Co-delivery of drugs and genetic materials was successfully attained through a carbonate apatite delivery device. On 4T1 cells, siRNAs against AKT and ERBB2 plus paclitaxel or docetaxel resulted in the largest increase in anti-cancer effects compared to CA/paclitaxel or CA/docetaxel. Therefore, these ingredients were selected for further in vivo investigations. Animals receiving injections of CA/paclitaxel or CA/docetaxel loaded with siRNAs against AKT and ERBB2 possessed significantly smaller tumors compared to CA/drug-treated mice. Interestingly, synergistic interactions in target protein knock down with combinations of CA/AKT/paclitaxel, CA/ERBB2/docetaxel were documented via western blotting.

## 1. Introduction

Combinations of individual therapeutics indicated against malignancies might result in superior outcomes in resolving the complexity of cancer cells. Among the tackled complexities is an acquired or intrinsic resistance of the cancer cells to treatment. In fact, combining different therapeutic strategies might provide various benefits as follows: (a) Maximized therapeutic efficacy without increased overall toxicity to the host due to different mechanisms of action; (b) prevention of the development of resistance to single agents; (c) covering the heterogeneous tumor cell population with different drug sensitivity profiles; and (d) possible synergy between therapeutics, resulting in increased anticancer efficacy [1].

The underlying mechanisms of resistance might involve genetic alterations, which could be depleted via RNA interference technologies, such as small interfering RNAs (siRNAs). Phenotype modifications through suppression of mRNA transcripts by the usage of siRNA, might render the cancer cells more responsive to the accompanying chemotherapeutic agents. In the co-delivery of nucleic acids and small molecule drugs, protective vectors would assist in boosted cellular uptake, lysosomal escape, and protection against serum nucleases together with limited off-target effects.

Nano-based formulations have the potential to be versatile and multifunctional, enabling the co-delivery of multiple agents entrapped in the nanoparticles’ structure to gain synergistic anticancer effects or multi-functions.

Co-delivery of drugs and siRNAs has been documented via application of different carrier systems [2,3,4]. In one study, doxorubicin and Bcl-2-targeted siRNA were applied via a polyethylenimine-coated graphene oxide (PEI-GO) vehicle. Here, the GO moiety adsorbs the therapeutics while PEI enhances cell membrane penetration [2]. In another example, a triblock polymer loaded with VEGF siRNA and paclitaxel delivered augmented efficacy against paclitaxel [3]. *N*-succinyl chitosan-poly-l-lysine-palmitic acid (NSC-PLL-PA) has been utilized in the synthesis of a triblock copolymer for co-delivery of P-glycoprotein-targeted siRNA and doxorubicin at pH 7.4. These therapeutics are released upon degradation of the carrier in the acidic pH of lysosomes [4].

In fact, numerous examples of synthetic dual siRNA and drug-delivery vehicles have been reported in the literature, with varying formulations and results [5,6,7,8]. However, carriers consisting of protein or peptide origins have demonstrated superiority due to biodegradability, complex structure details and functional groups, feasible adjustments in the synthesis process, and also compatibility in the construction of multifunctional hybrid materials [9]. 

As published data demonstrate, a lipoproteoplex has been developed for the purpose of dual delivery of siRNA and doxorubicin, with the ability to condense siRNA and encapsulate the small-molecule chemotherapeutic, doxorubicin. The lipoproteoplex demonstrates improved doxorubicin loading, resulting in a substantial decrease in MCF-7 cell viability, plus effective transfection of GAPDH (60% knockdown) in MCF-7 breast cancer cells [9].

Inorganic carbonate apatite is a recently developed nanocarrier synthesized via calcium phosphate precipitation in the presence of bicarbonate. The controlled crystal growth dynamic leads to the formation of particles with a size ranging between 50 and 300 nm. These carriers encompass optimal features of efficient endocytosis, fast dissolution rate in endosomal acidic pH, and effective release of loaded therapeutics. In more detail, addition of 3 mM of Ca in the synthesis of carbonate apatite results in the formation of nanoparticles with a size less than 50 nM. In presence of 10 nM of paclitaxel, NPs reach a maximum size of around 170 nm. Markedly, paclitaxel-loaded carbonate apatite demonstrates a 20.71 ± 4.34% loading efficiency [10]. Carbonate apatite nanoparticles have been utilized for co-delivery of anti-cancer drugs and various siRNAs, resulting in improved outcomes [11,12].

Members of the human epidermal growth factor receptor (HER) family are among the key factors in regulating the response of breast cancer cells to chemotherapy. As another mediator in the complex pathway of oncogenesis, AKT, a serine–threonine protein kinase, has been heavily studied.

AKT signaling has been shown to be regulated by enhanced HER signals, mutational activation of Ras leading to AKT activation via the PI3K pathway, or the mutational inactivation of the PTEN phosphatase resulting in diminished AKT activity. Secondary to these alterations, AKT kinase activity would be augmented. Thereafter, increased AKT kinase activity on its own and in the absence of any upregulation in AKT protein concentrations may have a broader effect on oncogenesis as well as the cellular response to cancer therapy [13].

Thus, various components of these inter-related signaling cascades have been targeted via siRNA application in this study to obtain an accurate assessment of their role in the response of the cancer cells to chemotherapy agents.

## 2. Materials and Methods

### 2.1. Materials

Dulbecco’s modified eagle medium (DMEM), calcium chloride dehydrate (CaCl_2_.2H_2_O), sodium bicarbonate (NaHCO_3_), dimethyl sulphoxide (DMSO), 3-(4,5-dimethylthiazol-2-yl)-2,5-diphenyl tetrazolium bromide (MTT), phosphoric acid solution (H_3_PO_4_), trifluroacetice acid (TFA; CF_3_COOH), Ethylene Diamine Tetraacetic Acid (EDTA), and anti-cancer drugs docetaxel (Doc) and paclitaxel (Pac) were purchased from Sigma-Aldrich (St Louis, MO, USA). DMEM powder, fetal bovine serum (FBS), trypsin ethylene diamine tetraacetate (trypsin-EDTA), 4-(2-hydroxyethyl)-1-piperazineethanesulfonic acid (HEPES), and penicillin-streptomycin were obtained from Gibco BRL (Carlsbad, CA, USA). All functionally validated siRNAs used in this study (listed in Table 1) were obtained from Qiagen and dissolved in RNase-free water provided by the company to obtain 10-μM stock solution. MCF-7, 4T1 and MDA-MB-231 cells were originally from ATCC.

### 2.2. Cell Culture

MCF-7 and MDA-MB-231 as human breast cancer cell lines and 4T1 as a mouse breast cancer cell line were cultured in 75-cm^2^ tissue culture flasks (Nunc, Orlando, FL, USA) using DMEM supplemented with 10% FBS, 1% penicillin and streptomycin, and 1% HEPES at 37 °C in a humidified 5% CO_2_-containing atmosphere.

### 2.3. Generation and Characterization of Carbonate Apatite

Carbonate apatite nanoparticles were manufactured via addition of Ca^2+^ from 1 M CaCl_2_ stock solution to the bicarbonate-buffered cell culture medium (DMEM, pH adjusted to 7.5, containing 44 mM HCO^3−^), which already contained the third reactant (0.9 mM phosphate), followed by incubation at 37 °C for 30 min. As a result, microscopically visible carbonate apatite particles were formed through precipitation following nucleation in a supersaturated solution. Turbidity determination and size and zeta potential measurement of the variously formulated nanoparticles were employed for characterization of the resulting products [10]. 

### 2.4. Complexation of Drugs and siRNAs with Carbonate Apatite

Various concentrations of drugs and siRNAs (Table 1) were added in DMEM media (44 mM bicarbonate, pH 7.5) containing particular concentrations of CaCl_2_ and subjected to a 30-min incubation at 37 °C to allow formation of complexes.

### 2.5. In Vitro Viability Assay

Cytotoxicity of carbonate apatite nanoparticles alone and also differently loaded NPs on human and murine breast cancer cell lines was assessed by MTT assay. Briefly, the cells from the exponential growth phase were seeded in 24-well plates (Griener, Frickenhausen, Germany) (approximately 50,000 cells/well) in DMEM with 10% FBS at 37 °C with 5% CO_2_. After 24 h, cells were exposed to various treatments for a consecutive period of 48 h. Two days later the viability was assessed by adding 50 μL MTT solution (5 mg/mL in phosphate buffered solution (PBS)) to each well and incubating for 4 h in dark. Then, the medium was removed and 300 μL DMSO was added to each well to dissolve the purple formazan crystals. Formazan quantification in the form of optical density (OD) was performed at test and reference wavelengths of 595 nm and 630 nm by a plate reader (benchmark plus, Bio Rad).

Cell viability was determined using the following formula:$$\mathrm{Cell}\;\mathrm{viability}\;(\%)\;(\mathrm{CV})\;=\;\frac{{\mathrm{OD}}_{\left(\mathrm{treated}\right)}-{\mathrm{OD}}_{\left(\mathrm{reference}\right)}}{{\mathrm{OD}}_{\left(\mathrm{untreated}\right)}-{\mathrm{OD}}_{\left(\mathrm{reference}\right)}}\;\times \;100.$$

The reference was the optical density of DMSO only in the applied wavelengths. 

Each experiment was done in triplicate and results are expressed as mean ± SD of % of cell viability. Subsequently, an increase in cytotoxic effect of the loaded NPs was calculated as follows:Increase in toxicity (%) = CV _baseline treatment_ – CV _complete treatment,_
where CV _baseline treatment_ and CV _complete treatment_ represent the cell viability resulting from the baseline treatment and complete treatment, respectively. In all bar charts (Appendix A) displaying cell viability values, each two adjacent bars were compared together and an increase in toxicity was calculated for all different concentrations and expressed as mean ± SD.

### 2.6. Sodium Dodecyl Sulfate Polyacrylamide Gel Electrophoresis (Sds-Page) and Western Blot

Treated cells in each well of a 24-well plate were lysed by addition of 200 μL of whole cell IP-lysis buffer and protein sample was collected. Then, 5 μL solution was used for estimation of total protein content using the BSA assay kit according to instruction provided by the manufacturer (Quick-start Bradford protein assay kit, Bio Rad, Hercules, CA, USA).

Samples of the cell lysate containing equal amounts of total protein (e.g., 10 μg) were mixed with 10 μL of 10× loading dye and heated for 5 min at 95 °C and then resolved by SDS-PAGE using stain free mini protean SFX gels (10 wells) in 1X running buffer. In total, 7 μL of precision plus protein standards-dual color were used as a molecular weight marker to confirm the molecular weight of the proteins in the samples. The protein samples were transferred from gel to the 0.2-μm polyvinylidene difluoride (PVDF) membranes and attached using a trans-blot turbo transfer system (Bio Rad). Membranes were blocked in 5% skimmed milk in 1X TBST for an hour at room temperature.

The membrane was probed with indicated primary antibody (Table 2) overnight at 4 °C. Unbound primary antibodies were washed using 1X TBST buffer for 5 times, 5 min each, with gentle agitation. Blots were probed with horseradish peroxide conjugated secondary antibody (anti-rabbit IgG, 1:3000) for 1 h at room temperature. TBST was again used to remove excess secondary antibody by 5 wash cycles each 5 min long with gentle agitation. Clarity Western Enhanced Chemiluminescence (ECL) substrate (Bio-Rad) was applied onto the membrane in the dark for 5 min and the signals on the membrane were visualized via a Bio-Rad Gel documentation system.

### 2.7. Formulation of Particles for In Vivo Study

The injectable nanoparticles were formulated in 100 μL of freshly prepared bicarbonated (44 mM) DMEM media to which CaCl_2_ was added. Samples were then incubated at 37 °C for 30 min followed by maintenance on ice to prevent aggregation during injection. In drugcontaining samples, 1.25 mg/kg of Pac and 1 mg/kg of Doc were used prior to incubation. In case of using siRNAs, 50 nM of each siRNA was added to the media prior to incubation. The resulting therapeutics were used for iv treatment of animals.

### 2.8. 4T1-Induced Breast Cancer Murine Model

Female Balb/c mice with the age of 6 to 8 weeks and body weight of 15 to 20 g were used in this study.

Animals were maintained in a 12:12 light:dark condition and provided with food ab libitum and water. All experiments were performed in complete adherence to the regulations of Monash University Animal Welfare Committee. The details of the animal study were approved by Monash Animal Ethics Committee on 3 August 2012 with the project identification code of MARP/2012/087 under the title of “Delivery of anti-cancer drugs to breast cancer cells using nanoparticles”. Tumor induction was performed via subcutaneous injection of 4T1 cells (in 100 μL PBS) on the mammary fat pad of mice. Injection day was considered as day 1 of the animal study. The development of tumor was regularly assessed through manual examination of the injection site. Randomization and treatment were carried out when the volume of the tumor reached an average of 13.20 ± 2.51 mm^3^ (Table 3). Treatment was administered via intravenous injection through the right or left caudal vein. Duration of study was 30 days, which involved close monitoring of the animals together with recording their body weights and tumor outgrowth every other day. The following formula was used for calculation of the tumor volume:$$\mathrm{Tumor}\;\mathrm{volume}\;({\mathrm{mm}}^{3})\;=\;\frac{\left(\mathrm{Length}\;\times {\mathrm{Width}}^{2}\right)}{2}$$

### 2.9. Statistics

For determining statistical significance of quantifications, student’s *t*-test was used; all data are presented as mean ± SD. Data was considered significant for *p* values < 0.05.

## 3. Results and Discussion

### 3.1. Cytotoxicity of siRNA-Loaded NPs on MCF-7 and 4T1 Cells

To explore the efficacy of carbonate apatite in the delivery of siRNAs into the cells and also evaluate the effect of single gene knock down on the viability of cancer cells, carbonate apatite nanoparticles harboring single siRNA were tested on MCF-7 and 4T1 cells. Here, 4 mM of CaCl_2_ was applied in the presence of various amounts of single siRNAs to produce treatment formulations.

Binding of siRNA to NPs, in the presence of increasing concentrations of calcium and fixed siRNA amounts, is saturable and reaches a plateau level, in spite of the escalating pattern of particle size. Higher amounts of calcium together with limited endogenous phosphate (0.9 mM) accounts for an increased particle size due to aggregation and not larger individual particles. Limited surface area of the particles secondary to aggregation results in restricted siRNA binding [14].

Remarkably, CA capacity in the efficient delivery of siRNAs into their action site is displayed by the considerable enhancement in the cytotoxic effects for almost all amounts of siRNAs complexed with CA compared to free siRNA. The most killing effect on MCF-7 and 4T1 seems to be achieved by muting the HER2 or ROS1 cascade (Table 4) (Appendix A), indicating the strong impact of ERBB2 or ROS signaling on the survival of both cell lines.

In another research, carbonate apatite nanoparticles have been confirmed as efficient carriers for electrostatically associated siRNA inside the cells, releasing bound siRNA from endosomes to the cytosol through pH-responsive self-dissolution. This was concluded based on the significant increase in the fluorescence intensity of the intracellular components from the cells treated with CA/fluorescent siRNA compared to untreated and CA-treated cells [14].

Moreover, application of different concentrations of ‘Allstars Negative Control siRNA’ (1 pM to 10 nM) loaded into a carbonate apatite structure resulted in no alteration of breast cancer cells’ viability [11].

However, the duration of response of the cancer cells to single-oncogene inhibitors, even the most potent oncogenes, is mostly short due to the development of resistance or upregulation of compensatory mechanisms in survival and proliferation signaling. Therefore, sustained and significant outcomes are mainly achieved with combinations of various classes of anti-cancer therapies. In case of no initial response to treatment or development of drug resistance, combinatory regimens are of great benefit to induce therapeutic response. In fact, putting off the causal cascades of resistance by one agent might result in re-sensitization of the cells to other therapeutics in the combination.

Markedly, pharmacokinetics and pharmacodynamics interactions between the agents in combination therapy might develop in case of a shared action site, uptake, and metabolism or elimination routes.

Delivery of drugs and siRNAs by means of carbonate apatite nanoparticles into cells was further studied. The aim was to evaluate the impact of silencing different signaling pathways on the response of cancer cells to the cytotoxic effects of drugs. Different concentrations of paclitaxel, docetaxel, mitomycin C, and topotecan (10 pM, 100 pM, and 1 nM) were used together with 1 pM of siRNAs against AKT, ERBB2, MAPK, and ROS1 and 3 mM Ca. Based on extensive experiments and with the aim of exploring possible synergistic effects, the lowest effective doses of therapeutics were applied. This was to perhaps achieve a therapeutic effect with lower doses and hence less side effects.

Notably, application of anti HER2 siRNA together with any of the four drugs caused positive changes in cytotoxicity against 4T1 cells (Appendix A). Additionally, co-delivery of Pac and ROS1 siRNA or Doc and MAPK siRNA seemed to sensitize 4T1 cells to the drug to a great extent.

Response of MDA-MB-231 cells to chemotherapy in case of single pathway silencing was also explored. Paclitaxel and docetaxel were used in complexes of carbonate apatite using 3 mM Ca and 10 pM, 100 pM, and 1 nM of the drug together with 1 nM of each siRNA. As the cell viability data reveals, Pac efficacy benefits from silencing ERBB2 or ROS1 pathways in terms of a highest increase in cytotoxicity on MDA-MB-231 cells (Appendix A). Whereas, AKT or ERBB2 pathways display a significant role in response to docetaxel in MDA-MB-231 cells, as with knock down of these two oncogenes, all concentrations of Doc exert a substantial higher cytotoxicity (Appendix A).

As the next step, simultaneous silencing of AKT and ERBB2 oncogenes was implemented to check if these pathways have any impact on the response to classical anti-cancer drugs.

Carbonate apatite was formed with 3 mM of CaCl_2_ and 1 pM of AKT and ERBB2 siRNA together with 10 pM, 100 pM, and 1 nM of paclitaxel, docetaxel, mitomycin C, and topotecan. Based on cell viability results (Appendix A), knock down of AKT and ERBB2 seems to have the greatest impact on the response to paclitaxel or docetaxel in 4T1 cells. The highest enhancement in cytotoxicity on 4T1 is equal to 19.97 ± 1.73% for Pac and 15.16 ± 3.55% for Doc, resulting from simultaneous blockade of AKT and ERBB2 signaling. Thus, these combinations were applied for in vivo studies since the animal tumor model was developed by local injection of 4T1 cells in mice.

On MDA-MB-231 cells, application of siRNAs for simultaneous silencing of two pathways led to augmented cytotoxicity of paclitaxel as well. This effect was higher with usage of AKT and ROS1 siRNA or co-delivery of AKT and MAPK siRNAs in the presence of Pac (Appendix A).

Table 5, Table 6 and Table 7 demonstrate the enhancement of the cytotoxic effect of classical anti-cancer drugs resulting from the silencing of various pathways in MCF-7, 4T1, and MDA-MB-231 cells, respectively.

According to the cell viability data displayed in Table 5, silencing AKT or ERBB2 regulates the response of MCF-7 cells to paclitaxel or docetaxel substantially. Thus, the effect of these therapeutic agents on the protein expression was further evaluated via Western blotting.

Induction of cell death followed by application of therapeutically loaded carbonate apatite could be associated with involvement of the apoptotic pathway, based on escalation of caspase-7-mediated signaling [12].

Silencing of AKT or HER2 signaling to expedite treatment outcomes in cancer is backed up by a solid body of knowledge. Expression of HER2 in MCF7 cells has been shown to increase under certain conditions [15,16].

According to recent research, docetaxel induces either apoptosis accompanied by survivin upregulation or necrosis and a lower rate of survivin upregulation in various breast cancer cell lines, based on cells receptor expression profile and the molecular phenotype. Additionally, inhibition of p-AKT was shown to revert survivin upregulation and also induce docetaxel-dependent apoptosis [17].

In another study, inhibition of NRF2 in ERBB2-overexpressing ovarian carcinoma cells was shown to suppress ERBB2 expression, which led to a decrease in phospho-AKT and enhanced p27 protein together with increased sensitivity of these cells to docetaxel cytotoxicity and apoptosis [18].

There are also other published data associating increased paclitaxel-induced apoptosis with inhibition of AKT in ovarian cancer cells. While paclitaxel-resistant cells demonstrated higher levels of p-AKT compared to paclitaxel-sensitive cells, inhibition of AKT increased paclitaxel therapeutic efficacy in both cell lines [19].

Moreover, combination of shRNA against AKT1 with paclitaxel exerted synergistic anti-cancer effects, thus inhibiting the growth of human breast cancer MDA-MB-231 and MCF-7 cells, and breast cancer MDA-MB-231 cell xenografts in mice as well. The combination therapy demonstrated enhanced anti-cancer effects through inhibition of AKT1 signaling and induction of apoptosis [20].

Extensive in vitro and in vivo research has associated ERBB2 overexpression with resistance of breast cancer cells to certain chemotherapeutic agents, namely docetaxel and paclitaxel. Furthermore, in clinical studies, herceptin, the anti-ERBB2 antibody, enhanced the antitumor activity of paclitaxel and doxorubicin against ERBB2-overexpressing human breast cancer xenografts, and the paclitaxel response rate of patients with ERBB2-overexpressing breast cancers was significantly higher among patients receiving paclitaxel plus herceptin than those receiving paclitaxel alone [21]. Additionally, co-administration of a PI3K inhibitor with cytotoxic or targeted anticancer agents, such as carboplatin, paclitaxel, or erlotinib, led to increased tumor growth inhibition over the corresponding single agents [22].

Moreover, high AKT activity has been shown to be responsible for the enhanced resistance of ERBB2-overexpressing cancer cells toward chemotherapeutic agents, and inhibition of AKT activation by peptide aptamer resulted in the restoration of regular sensitivity of breast cancer MCF7 cells towards paclitaxel [23].

In another research, siRNA-based knockdown of HER-2 conferred increased sensitivity to paclitaxel in endometrial cancer cells, attenuating the induction of p-AKT on paclitaxel stimulation, which was cancelled by inactivating AKT by the introduction of a dominant-negative form [24].

### 3.2. Effect of CA/siRNA/Drug at the Protein Level

As cell viability assays revealed, silencing AKT or ERBB2 exerts the maximum effect on the enhancement of cytotoxicity of Pac or Doc on MCF-7. Thus, the underlying mechanism of reduced viability following administration of these treatments was investigated via western blotting. This was performed to confirm that the applied siRNA is effective in the knock down of the target protein synthesis and also to observe the changes in protein expression leading to enhanced chemo sensitivity.

AKT expression following treatment of MCF-7 cells with free AKT siRNA and CA-complexed AKT siRNA was assessed and compared with untreated cells. As the bands on the blot and densitometry analysis reveal (Figure 1), complexation of siRNA with carbonate apatite leads to significantly decreased expression of AKT compared to free AKT siRNA and untreated cells (*p* value < 0.05). This confirms the efficacy of CA in the successful delivery of siRNA and knock down of the target protein.

The same experiment design was utilized with MAPK siRNA. According to Figure 2, free MAPK siRNA does not change the expression level of MAPK protein whereas CA/MAPK significantly knocks down production of the target protein (*p* value < 0.05).

Next, changes in protein expression resulting from the combinations of drugs and siRNAs, as highlighted in the cell viability data, were studied. MCF-7 cell lysate treated with free and CA-bound AKT siRNA, CA/AKT/Pac and CA/Pac were loaded on the gels. According to the resulting bands (Figure 3), the lowest level of AKT protein was obtained by CA/AKT/Pac treatment versus CA/Pac or CA/AKT. This implies the development of synergistic interactions to suppress survival pathways in the presence of classical anti-cancer drugs together with genetic downregulation of survival proteins, which in turn might lead to enhanced sensitivity of the cells to the anti-cancer medication.

The same synergistic pattern, but with a lower potency, was observed with application of CA/AKT/Doc in the downregulation of AKT protein against CA/AKT or CA/Doc (Figure 4).

Based on a substantial enhancement in efficacy, the impact of different combinations of ERBB2 siRNA with paclitaxel or docetaxel on the expression level of ERBB2 protein was also examined. As the blot in Figure 5 shows, the highest knock down was achieved by co-delivery of CA-bound Doc and ERBB2 siRNA followed by CA/Pac/ERBB2. Again, the synergy in the therapeutic efficacy of the drug together with siRNA is present since the impact of the drug or siRNA as a single agent is considerably weaker.

### 3.3. In Vivo Efficacy of CA/Drug/siRNA

According to cell viability data on 4T1 cells and the enhancement in the cytotoxicity of drugs, co-delivery of AKT and ERBB2 siRNAs together with Pac or Doc resulted in the highest increase in the drug’s efficacy. Thus, these two combinations were applied in animal study.

For comparison of the in vivo efficacy of the therapeutics in the first batch, animals were treated on day 8 and 11. Formulations were prepared by mixing 4 mM Ca, 1.25 mg/kg Pac, 50 nM AKT siRNA plus 50 nM ERBB2 siRNA in 100 µL of HCO_3_-DMEM. There were no significant changes in the pattern of body weight change and also general signs and symptoms of the animals among different groups of the studies.

As the t-test results reveal, CA-AKT-ERBB2-Pac treatment significantly reduced the tumor volume on day 14 and 16 compared to Ca/Pac. Moreover, the group treated with CA/ERBB2 displayed significantly smaller tumors compared to the CA group on day 12 and 14. The effect of Pac complexed with CA on tumor regression was significant on day 14 versus free paclitaxel therapy (Figure 6).

In another batch, injections on day 10 and 13 encompassed preparations of 4 mM Ca, 1 mg/kg Doc, 50 nM AKT siRNA, and 50 nM ERBB2 siRNA in 100 µL bicarbonated DMEM (Figure 7).

According to the *p* values calculated in the t-test, treatment of animals with CA/Doc/AKT/ERBB2 resulted in significantly smaller tumor volumes on days 16, 18, 20, and 22 compared to CA/Doc. Whereas CA/ERBB2 displayed a significant anti-tumor efficacy only on day 16 compared to CA, and CA/AKT was not effective in tumor regression. Animals treated with CA/Doc had significantly smaller tumors on day 18 and 20 compared to Doc-treated animals.

ERBB2 overexpression has been linked to elevated levels of inhibitory phosphorylation of Cdc2 and suppression of paclitaxel-induced cell death in breast cancer cells via deregulation of the G2/M cell cycle checkpoint. This provides a mechanistic rationale for the association between ERBB2 overexpression and paclitaxel resistance. Interestingly, overexpression of a subunit of PI3k in ovarian cancer cells has been shown to confer paclitaxel resistance. Additionally, selective inhibition of the PI3k pathway could restore the efficacy of paclitaxel in those cells [25]. Thus, alterations in the expression and activity levels of key components of these signaling networks regulating cellular proliferation and survival may confer paclitaxel resistance. Further, circumvention of this type of resistance might be achieved via application of selective inhibitors of these proteins to increase drug sensitivity.

Simultaneous targeting of AKT and ERBB2 to achieve improvements in eradication of oncogenesis has been brought up in various studies.

Overexpression or activation of HER members and also AKT has been linked to limited benefits of treatment in patients, leading to poor prognosis in breast cancer. In fact, functionally relevant alterations in AKT1 could be a putative mediator of tumor progression and drug resistance. Moreover, amplification or gain-of-function mutations of ERBB2 can account for hyperactivation of the AKT cascade in breast cancer cells. Crosstalk with heterologous receptors and amplification of HER2 signaling, amplifications of the PI3K/AKT pathway, and de-repression and/or activation of compensatory survival pathways through increased PI3K/AKT signaling are among the resistance mechanisms against anti-HER2 therapy. Factors associated with resistance to ERBB2-targeted agents have been invariably associated with a reactivation of the PI3K/AKT signaling cascade [26,27]. However, defects in HER2/PI3K/AKT axes and their impact on regulation of the response of cancer cells to treatment need further investigation.

In an experimental design, HER2-positive MCF7 cells showed a PI3K-dependent increase in AKT activity together with increased resistance to a panel of five chemotherapeutic agents with known different mechanisms of action (paclitaxel, doxorubicin, 5-fluorouracil, etoposide, and camptothecin). Selective inhibition of PI3K or AKT in these HER2-overexpressing MCF7 cells reduced the levels of phosphorylated (activated) AKT and sensitized the cells to the chemotherapeutic agents. It has been further confirmed that expression of a constitutively active AKT vector alone in MCF7 cells caused similarly increased resistance of the cells to the chemotherapeutic agents [13]. Therefore, the HER2/PI3K/AKT pathway is confirmed to play a causal role in the resistance of breast cancer cells against several therapeutics, and targeting components of this pathway might resolve the resistance and enhance the efficacy of the treatments. In view of that, the in vitro and in vivo results of this study are in alignment with the documented role of AKT and ERBB2 in drug resistance.

Strikingly, the efficacy of simultaneous knock down of AKT and ERBB2 in augmenting the response of cancer cells to docetaxel was reported for the first time in this study.

Markedly, the strategic location of AKT and its activation by multiple upstream signal transduction pathways makes it a better target than its upstream targets, such as HER2, Ras, or PI3K, in sensitizing cancer cells to chemotherapy or radiotherapy. Thus, there could be clinical benefits from an appropriate combination of conventional chemotherapeutic drugs with a new generation of signal transduction inhibitors that inhibit the HER/PI3K/AKT pathway for the treatment of breast cancer.

A summary of the ERBB2 and AKT signaling cascades regulating cell’s proliferation, survival, and resistance to apoptosis and treatment is illustrated in Figure 8. This is for clarification of the enhanced efficacy of Pac or Doc attained by simultaneous blockade of AKT and ERBb2 cascades and the impact on the response of the cells to the drugs.

As revealed through the extensive experiments on breast cancer cells and animal models plus protein expression studies, simultaneous knock down of AKT1 and ERBB2 expression in the presence of paclitaxel or docetaxel leads to a substantial increase in cellular response to the chemotherapeutic agent both in culture and an animal model. Synergistic interactions in target protein knock down have been documented via western blotting in the cells treated with these combinations, which would shed light on the underlying mechanisms of the enhanced sensitivity of the cells to treatment. Taken together, with a favorable in vitro profile and also superior in vivo efficacy, co-administration of genetic materials and classical chemotherapeutics could propose a promising platform for improved strategies against cancer cells in upcoming practice.

## Figures and Tables

**Figure 1 pharmaceutics-11-00458-f001:**
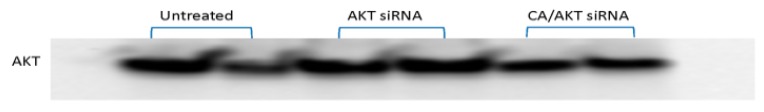
AKT protein expression in MCF-7 cells. Cells were treated with media (untreated), free AKT siRNA, and CA/AKT siRNA formed with 3 mM of CaCl_2_ and 1 nM siRNA for 44 h. Cellular lysates were resolved by SDS-PAGE and transferred in PVDF membrane followed by incubation with primary antibodies raised in rabbit against AKT. HRP-conjugated goat anti-rabbit secondary antibody was used to detect the chemiluminescent signals.

**Figure 2 pharmaceutics-11-00458-f002:**
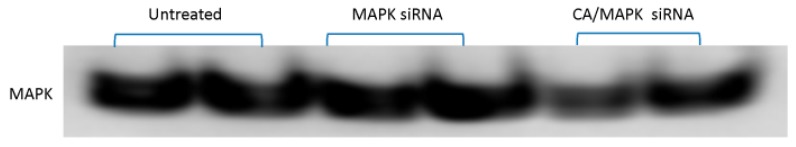
MAPK protein expression in MCF-7 cells. Cells were treated with media (untreated), free MAPK siRNA, and CA/MAPK siRNA formed with 3 mM of CaCl_2_ and 1 nM siRNA for 44 h. Cellular lysates were resolved by SDS-PAGE and transferred into PVDF membrane followed by incubation with primary antibodies raised in rabbit against AKT. HRP-conjugated goat anti-rabbit secondary antibody was used to detect the chemiluminescent signals.

**Figure 3 pharmaceutics-11-00458-f003:**
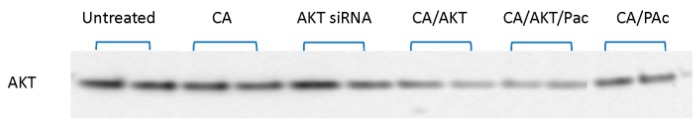
AKT protein expression in MCF-7 cells. Cells were treated with media (untreated), carbonate apatite, free AKT siRNA, CA/AKT siRNA, CA/AKT/Pac, and CA/Pac. Ingredients include 3 mM of CaCl_2_, 1 nM siRNA, and 1 nM Pac. Cellular lysates were run in SDS-PAGE and transferred into PVDF membrane and incubated with primary antibodies raised in rabbit against AKT. HRP-conjugated goat anti-rabbit secondary antibody was used to detect the chemiluminescent signals.

**Figure 4 pharmaceutics-11-00458-f004:**
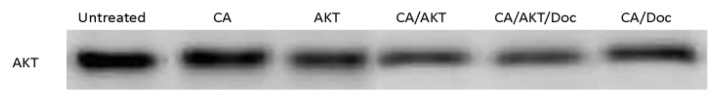
AKT protein expression in MCF-7 cells. Cells were treated with media (untreated), carbonate apatite, free AKT siRNA, CA/AKT siRNA, CA/AKT/Doc, and CA/Doc. Ingredients include 3 mM of CaCl_2_, 1 nM siRNA, and 1 nM Doc. Cellular lysates were run in SDS-PAGE and transferred into PVDF membrane and incubated with primary antibodies raised in rabbit against AKT. HRP-conjugated goat anti-rabbit secondary antibody was used to detect the chemiluminescent signals.

**Figure 5 pharmaceutics-11-00458-f005:**
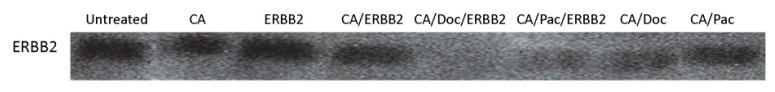
ERBB2 protein expression in MCF-7 cells. Cells were treated with media (untreated), carbonate apatite, free ERBB2 siRNA, CA/ERBB2 siRNA, CA/Doc/ERBB2, CA/Pac/ERBB2, CA/Doc, and CA/Pac. Ingredients include 3 mM of CaCl_2_, 1 nM siRNA, and 1 nM drug. Cellular lysates were run in SDS-PAGE and transferred into PVDF membrane and incubated with primary antibodies raised in rabbit against ERBB2. HRP-conjugated goat anti-rabbit secondary antibody was used to detect the chemiluminescent signals.

**Figure 6 pharmaceutics-11-00458-f006:**
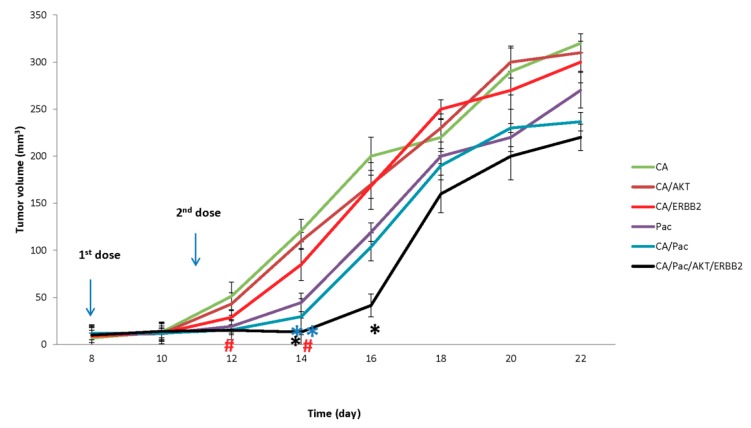
Effect of silencing AKT and ERBB2 pathways on in vivo efficacy of paclitaxel. Mice were purchased from Razi Research Institute, Tehran, Iran. Approximately 10^6^ 4T1 cells were inoculated subcutaneously on the mammary pad of mice. Based on tumor volume calculations, mice were randomized and treated intravenously through tail-vein injection on day 8 and 11. The therapeutics included 100 μL of carbonate apatite entailing 4 μL of 1M CaCl_2_ and 50 nM of AKT and ERBB2 siRNA plus 1.25 mg/kg Pac. Body weight and tumor outgrowth were monitored every other day. Data is represented as mean ± SD, *n* = 6 and values are significant when * *p* value < 0.05 for CA/Pac/AKT/ERBB2 vs. CA/Pac, # *p* value < 0.05 for CA/ERBB2 compared to CA and ** *p* value < 0.05 for CA/Pac against Pac.

**Figure 7 pharmaceutics-11-00458-f007:**
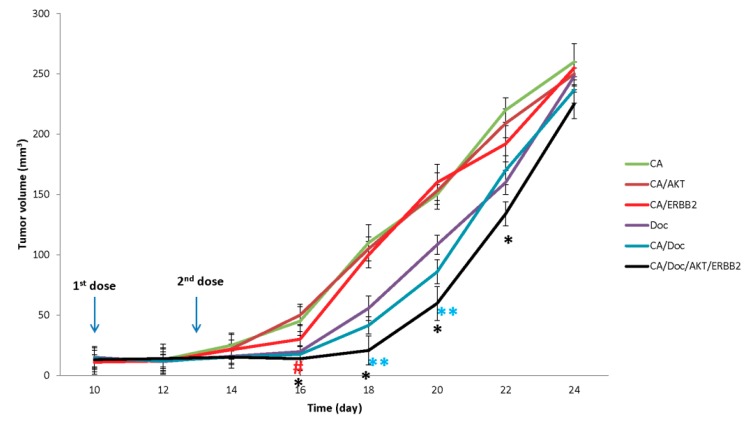
Effect of silencing AKT and ERBB2 pathways on in vivo efficacy of docetaxel. Mice were purchased from Razi Research Institute, Tehran, Iran. Approximately 10^6^ 4T1 cells were inoculated subcutaneously on the mammary pad of mice. Based on tumor volume calculations, mice were randomized and treated intravenously through tail-vein injection on day 10 and 13. The therapeutics included 100 μL of carbonate apatite entailing 4 μL of 1M CaCl_2_ and 50 nM of AKT and ERBB2 siRNA plus 1mg/kg Doc. Body weight and tumor outgrowth were monitored every other day. Data is represented as mean ± SD, *n* = 6 and values are significant when * *p* value < 0.05 for CA/Doc/AKT/ERBB2 vs CA/Doc, # *p* value < 0.05 for CA/ERBB2 compared to CA and ** *p* value < 0.05 for Ca/Doc against Doc.

**Figure 8 pharmaceutics-11-00458-f008:**
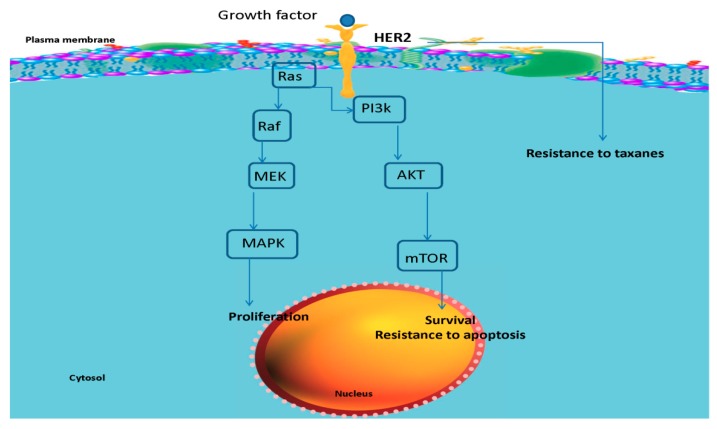
Summary of the ERBB2 and AKT signaling cascades regulating the cell’s proliferation, survival, and resistance to apoptosis and treatment.

**Table 1 pharmaceutics-11-00458-t001:** List of validated siRNAs (Qiagen, MD, USA) that were used in this study.

siRNA	Target Sequence	Targeted Gene	Validation Cell Line	% knockdown
Hs-AKT1-5	AATCACACCACCTGACCAAGA	AKT (protein kinase B)	HeLa S3	90
Hs_MAPK1_l0	AAGTTCGAGTAGCTATCAAGA	Mitogen-activated protein kinase	HeLa S3	90
Hs-ROS1-5	AAGGTAATTGCTCTAACTTTA	ROS Proto-Oncogene 1, Receptor tyrosine kinase	HeLa	87
Hs-ERBB2-14	AACAAAGAAATCTTAGACGAA	Receptor tyrosine-protein kinase ERBB-2 or human epidermal growth factor receptor 2 (HER2)	MCF-7	92

**Table 2 pharmaceutics-11-00458-t002:** Information for the primary antibodies used for western blot in this study.

Name	Manufacturer	Molecular Weight	Clonality	Used Dilution
AKT (pan) (C67E7) Rabbit mAb	Cell signaling (Danvers, MA, USA)	60 KDa	Monoclonal	1:1000
p 44/42 MAPK (Erk 1/2) (137F5) Rabbit mAb	Cell signaling (Danvers, MA, USA)	42/44 KDa	Monoclonal	1:1000
ERBB2 (HER2)	Thermo Fisher scientific (Waltham, MA, USA)	200 KDa	Polyclonal	1:1000
GAPDH (glyceraldehyde-3-phosphate dehydrogenase)	Cell signaling (Danvers, MA, USA)	37 KDa	Monoclonal	1:3000

**Table 3 pharmaceutics-11-00458-t003:** Treatment groups used for in vivo study.

Group	Regimen
Untreated	-
CA	7 µL of 1 M CaCl_2_ in DMEM
CA	4 µL of 1 M CaCl_2_ in DMEM
Pac	1.25 mg/kg paclitaxel in DMEM
Doc	1 mg/kg docetaxel in DMEM
CA/Pac	1.25 mg/kg paclitaxel and 7 µL of 1 M CaCl_2_ in DMEM
CA/Pac	1.25 mg/kg paclitaxel and 4 µL of 1 M CaCl_2_ in DMEM
CA/Doc	1 mg/kg docetaxel and 4 µL of 1 M CaCl_2_ in DMEM
CA/AKT	50 nM of AKT siRNA and 4 µL of 1 M CaCl_2_ in DMEM
CA/ERBB2	50 nM of ERBB2 siRNA and 4 µL of 1 M CaCl_2_ in DMEM
CA/AKT/ERBB2/Pac	1.25 mg/kg paclitaxel and 50 nM of ERBB2 and AKT siRNA and 4 µL of 1 M CaCl_2_ in DMEM
CA/AKT/ERBB2/Doc	1 mg/kg docetaxel and 50 nM of ERBB2 and AKT siRNA and 4 µL of 1 M CaCl_2_ in DMEM

**Table 4 pharmaceutics-11-00458-t004:** Cytotoxicity (%) enhancement on 4T1 and MCF-7 cells treated with various siRNAs (1 pM, 10 pM, 100 pM, 1 nM, and 10 nM) incorporated into an apatite structure formed with 3 mM of CaCl_2_. The data is presented as mean ± SD compared to free siRNA.

Cell line	Treatment	siRNA Concentration				
		1 pM	10 pM	100 pM	1 nM	10 nM
4T1	CA/ERBB2	16.44 ± 5.77	33.24 ± 9.14	26.70 ± 3.93	28.09 ± 6.28	19.80 ± 1.13
	CA/AKT	14.11 ± 5.41	−4.58 ± 5.01	3.42 ± 1.23	−3.27 ± 4.48	−3.96 ± 2.98
	CA/MAPK	10.68 ± 3.69	−0.72 ± 0.87	17.59 ± 1.04	5.36 ± 1.15	25.45 ± 4.64
	CA/ROS1	13.85 ± 6.99	14.68 ± 3.87	27.46 ± 1.17	16.65 ± 1.44	35.56 ± 2.81
MCF-7	CA/ERBB2	07.49 ± 8.01	24.56 ± 3.19	13.82 ± 3.34	19.29 ± 3.91	8.63 ± 1.67
	CA/ROS1	12.52 ± 2.79	13.69 ± 3.07	23.97 ± 2.13	28.88 ± 2.29	29.70 ± 3.96

**Table 5 pharmaceutics-11-00458-t005:** Effect of silencing various pathways on cytotoxicity of classical anti-cancer drugs in MCF-7 cells. Data is presented as mean ± SD compared to CA/drug.

Treatment on MCF-7 Cells	Drug Concentration		
	10 pM	100 pM	1 nM
CA/Pac/AKT	10.93 ± 1.05	8.33 ± 0.84	3.31 ± 0.45
CA/Pac/ERBB2	7.25 ± 1.91	4.11 ± 0.35	3.40 ± 1.81
CA/Pac/MAPK	−10.79 ± 2.12	1.53 ± 0.55	2.06 ± 0.35
CA/Pac/ROS1	4.19 ± 1.41	3.14 ± 0.74	3.64 ± 0.19
CA/Doc/AKT	5.14 ± 1.22	10.45 ± 1.33	11.69 ± 2.18
CA/Doc/ERBB2	8.95 ± 2.28	10.25 ± 0.99	14.28 ± 1.04
CA/Doc/MAPK	3.22 ± 0.91	4.27 ± 0.08	3.67 ± 1.72
CA/Doc/ROS1	1.06 ± 0.59	2.83 ± 0.71	1.25 ± 0.53

**Table 6 pharmaceutics-11-00458-t006:** Effect of silencing various pathways on cytotoxicity of classical anti-cancer drugs in 4T1 cells. Data is presented as mean ± SD compared to CA/drug.

Treatment on 4T1 Cells	Drug Concentration		
	10 pM	100 pM	1 nM
CA/Pac/AKT	−15.93 ± 0.95	8.70 ± 2.14	−2.68 ± 0.54
CA/Pac/ERBB2	12.93 ± 0.36	9.15 ± 0.14	2.42 ± 2.84
CA/Pac/MAPK	−13.79 ± 1.18	2.53 ± 0.20	2.16 ± 0.11
CA/Pac/ROS1	10.09 ± 1.03	2.04 ± 0.85	7.24 ± 0.07
CA/Doc/AKT	−5.04 ± 0.77	−9.43 ± 0.51	1.30 ± 0.62
CA/Doc/ERBB2	5.85 ± 1.08	7.21 ± 1.17	2.20 ± 1.95
CA/Doc/MAPK	3.16 ± 0.21	5.07 ± 0.98	2.67 ± 0.76
CA/Doc/ROS1	2.76 ± 0.58	1.81 ± 0.97	0.05 ± 0.42
CA/Mito/AKT	0.20 ± 0.71	4.97 ± 1.15	1.92 ± 0.28
CA/Mito/ERBB2	2.13 ± 3.64	5.11 ± 0.85	1.71 ± 0.38
CA/Mito/MAPK	1.26 ± 1.23	0.23 ± 1.45	7.77 ± 0.95
CA/Mito/ROS1	0.72 ± 0.92	0.52 ± 0.16	4.20 ± 1.12
CA/Topo/AKT	0.02 ± 1.64	3.09 ± 0.42	5.28 ± 3.07
CA/Topo/ERBB2	4.54 ± 2.27	5.85 ± 0.47	0.86 ± 2.51
CA/Topo/MAPK	6.54 ± 0.55	0.55 ± 0.51	4.74 ± 0.88
CA/Topo/ROS1	0.58 ± 0.25	0.03 ± 0.36	0.52 ± 0.81
CA/Pac/AKT/ERBB2	19.97 ± 1.73	5.48 ± 2.09	11.63 ± 2.23
CA/Doc/AKT/ERBB2	7.87 ± 1.82	1.45 ± 0.37	15.16 ± 3.55
CA/Mito/AKT/ERBB2	6.04 ± 0.28	0.59 ± 0.47	4.60 ± 0.82
CA/Topo/AKT/ERBB2	−10.41 ± 0.66	−4.73 ± 1.52	−10.77 ± 0.71

**Table 7 pharmaceutics-11-00458-t007:** Effect of silencing various pathways on cytotoxicity of classical anti-cancer drugs in MDA-MB-231 cells. Data is presented as mean ± SD compared to CA/drug.

Treatment on MDA-MB-231 Cells	Drug Concentration		
	10 pM	100 pM	1 nM
CA/Pac/AKT	4.60 ± 1.53	−8.84 ± 0.92	7.55 ± 1.43
CA/Pac/ERBB2	12.03 ± 2.44	0.59 ± 1.32	12.03 ± 3.57
CA/Pac/MAPK	8.96 ± 2.16	0.35 ± 0.36	8.25 ± 1.28
CA/Pac/ROS1	11.08 ± 2.09	0.35 ± 0.09	7.90 ± 1.83
CA/Doc/AKT	3.66 ± 0.24	8.97 ± 2.94	16.16 ± 4.52
CA/Doc/ERBB2	20.05 ± 4.61	7.97 ± 3.06	8.02 ± 1.49
CA/Doc/MAPK	−1.06 ± 0.58	6.14 ± 2.73	6.49 ± 1.91
CA/Doc/ROS1	0.59 ± 0.34	10.74 ± 3.16	7.90 ± 1.52
CA/Pac/AKT/ERBB2	7.07 ± 2.22	3.71 ± 0.85	13.29 ± 1.12
CA/Pac/AKT/MAPK	18.62 ± 3.64	14.42 ± 2.14	3.37 ± 3.47
CA/Pac/AKT/ROS1	11.26 ± 1.05	−0.12 ± 0.93	22.40 ± 3.44
CA/Pac/ERBB2/MAPK	1.32 ± 0.07	−1.44 ± 0.12	12.22 ± 2.33
CA/Pac/ERBB2/ROS1	−0.48 ± 0.52	−9.70 ± 0.49	5.51 ± 1.86
CA/Pac/MAPK/ROS1	9.58 ± 1.24	−4.79 ± 0.93	6.23 ± 2.22

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
