# Peer review of "Intracellular Delivery of siRNAs Targeting AKT and ERBB2 Genes Enhances Chemosensitization of Breast Cancer Cells in a Culture and Animal Model"

_pharmaceutics, 2019, doi:10.3390/pharmaceutics11090458_

Round 1

Reviewer 1 Report

Fatemain and coworkers hypothesize that co-delivery of siRNAs targeting cancer-related signaling cascades with a therapeutic chemotherapy agent (like doxorubicin or a taxol-derivative) will produce synergistic effects to increase drug toxicity. To test this hypothesis, they prepared carbonate apatite nanoparticles adsorbed with different gene-specific siRNAs (AKT, MAPK, ERBB2) and/or chemotherapeutics (paclitaxel or docetaxel) and tested their effect to inhibit growth in 2 human breast cancer cell lines (MCF-7 and 4T1) in vitro and in vivo. While co-delivery of siRNAs and chemotherapy agents is not a particularly novel concept, the authors present evidence of a synergistic effect of co-delivery with their carbonate apatite nanoparticle system with slight improvements to reduce tumor growth.

Major Concerns:

1)     Given the author’s expertise and previous publication record working with carbonate apatite nanoparticles, there is little introduction to this particular system while several other nanoparticle platforms are mentioned in the introduction. A brief section describing the performance of the carbonate apatite system will help inform the reader of the prior work in the area.

2)     Figures 1-10, no indications of statistical tests are identified, nor lines to indicate which samples would be compared (i.e., treated vs untreated, CA vs CA/siRNA, free siRNA vs CA/siRNA, etc.)

3)     Toxicity of treatments was only measured using an MTT assay, which will only measure cell viability. The decrease in cell viability may be the result of slowed cell growth compared to controls. Is there actually an increase in apoptotic cells?

4)     Does knockdown of each gene by transfection of siRNAs (as compared to using the nanoparticles) result in increased cell death when compared to a scrambled control sequence? Alternatively, does treatment with CA/scrambled siRNA result in no increased cell death?

5)     Table 4 seems unneeded and repeats the same data as demonstrated in Figure 1 and Figure 2.

6)     It is not clear why 1 pM of each siRNA was used for experiments performed in Figure 3-6 when 10 or 100 pM clearly produced a more significant effect on decreased viability in Figures 1 and 2.

7)     Given the low dosages of chemotherapy used in Figures 3-6 (most of these compounds have reported IC50 in nM or µM range), it is not clear if the synergistic effect of co-delivery causes a significant change in drug IC50 or may be due to increased uptake of the carrier system.

8)     Again, Tables 5-8 seem unneeded and repeats the same data presented in Figures 3-6

9)     In the in vivo experiments, were there any changes in body weight or were mice monitored for signs of non-specific toxicities associated with the broad distribution of particles?

10)  Were tumors histologically evaluated for overall increases in cell death or increased apoptosis?

Minor Concerns:

1)     The manuscript should be reviewed for minor English grammatical issues to improve clarity and readability. For example in line 28, the sentence is very difficult to read.

2)     Line 68 – “PI-3K” should be corrected to PI3K pathway.

3)     Line 79 – chemical formula style needs to be corrected. The dot before the hydrate appears to be a subscript or a period and should be corrected as a dot. H2O should be H2O with a correct subscript. (CaCl2•2H2O)

4)     Line 81 – similarly, H3PO4 needs to be formatted with the correct chemical notation including subscripts (H3PO4)

5)     Line 82 – similarly, CF3COOH needs to be formatted in the correct chemical notation including subscripts (CF3COOH)

6)     Line 94 – Formatting of heading should be consistent with rest of manuscript (sentence case). “Carbonate Apatite” should be lowercase.

7)     Line 129 – Centrifugation speed should ideally be reported in rcf or ×g.

8)     Line 135 – Electrophoresis conditions are commonly reported in voltage not current. Was this experiment run under constant voltage or constant current conditions?

9)     Line 138 – Similar to above, electrophoresis conditions are commonly reported in voltage not current. Was the experiment run under constant voltage or constant current?

10)  Line 152 – Kg should be lowercase (i.e., 1.25 mg/kg of Pac)

11)  Line 169 – Title in table should have “in vivo” in italics (i.e., in vivo)

12)  Line 169 – Inconsistent, long spacing in line CA/AKT/ERBB2/Doc (1          mg/kg docetaxel). May be an issue with PDF generation.

13)  Line 298 – Drug concentration (10 PM) should be corrected to “10 pM”

14)  Line 300 – Drug concentration (10 PM) should be corrected to “10 pM”

15)  Line 322 – “in vitro” and “in vivo” should be italicized

16)  Figure 12 and Figure 13 should be combined as a multi-panel figure as they describe the same data.

17)  Similarly, Figure 14 and relevant portions of Figure 16 should be combined as a multi-panel figure as they describe the same data.

18)  Figure 15 and relevant portions of Figure 16 should be combined as a multi-panel figure as they describe the same data.

19)  Line 401 – References to “Figure 45” and “Figure 46” should be updated appropriately.

20)  Figure 17 and Figure 18 should be combined as a multi-panel figure as they describe the same data.

21)  Line 507 – there should be consistency in the manuscript with capitalization of Western blotting.

22)  Line 509 – “in vitro” and “in vivo” should be italicized.

Author Response

Reviewer 1

Fatemain and coworkers hypothesize that co-delivery of siRNAs targeting cancer-related signaling cascades with a therapeutic chemotherapy agent (like doxorubicin or a taxol-derivative) will produce synergistic effects to increase drug toxicity. To test this hypothesis, they prepared carbonate apatite nanoparticles adsorbed with different gene-specific siRNAs (AKT, MAPK, ERBB2) and/or chemotherapeutics (paclitaxel or docetaxel) and tested their effect to inhibit growth in 2 human breast cancer cell lines (MCF-7 and 4T1) in vitro and in vivo. While co-delivery of siRNAs and chemotherapy agents is not a particularly novel concept, the authors present evidence of a synergistic effect of co-delivery with their carbonate apatite nanoparticle system with slight improvements to reduce tumor growth.

Major Concerns:

1)        Given the author’s expertise and previous publication record working with carbonate apatite nanoparticles, there is little introduction to this particular system while several other nanoparticle platforms are mentioned in the introduction. A brief section describing the performance of the carbonate apatite system will help inform the reader of the prior work in the area.

Answer: Details on the characteristics of carbonate apatite nanoparticles were added in line 64- 73.

“Inorganic carbonate apatite is a recently developed nanocarrier synthesized via calcium phosphate precipitation in presence of bicarbonate. The controlled crystal growth dynamic leads to formation of particles with the size ranging between 50-300 nm. These carriers encompass optimal features of efficient endocytosis, fast dissolution rate in endosomal acidic pH and effective release of loaded therapeutics. In more detail, addition of 3 mM of Ca in the synthesis of carbonate apatite results in formation of nanoparticles with the size of less than 50 nM. In presence of 10 nM paclitaxel, NPs reach the maximum size of around 170 nm. Markedly, paclitaxel loaded carbonate apatite demonstrates 20.71 ± 4.34 % loading efficiency (10). Carbonate apatite nanoparticles have been utilized for co-delivery of anti-cancer drugs and various siRNAs resulting in improved outcome (11, 12).”

2)        Figures 1-10, no indications of statistical tests are identified, nor lines to indicate which samples would be compared (i.e., treated vs untreated, CA vs CA/siRNA, free siRNA vs CA/siRNA, etc.)

Answer: For better presentation of the data, figure 1-10 were eliminated from the text and could be included as supplementary figures at the end of the manuscript. Additional explanation was added in line 135 for clarification of the comparisons.

“In all bar charts (supplementary figures) displaying cell viability values, each two adjacent bars have been compared together and increase in toxicity was calculated for all different concentrations and expressed as mean± SD.”

3)        Toxicity of treatments was only measured using an MTT assay, which will only measure cell viability. The decrease in cell viability may be the result of slowed cell growth compared to controls. Is there actually an increase in apoptotic cells?

Answer: Information added in line 264.

“Induction of cell death followed by application of therapeutically loaded carbonate apatite could be associated with involvement of apoptotic pathway, based on escalation of caspase-7-mediated signal.”

4)        Does knockdown of each gene by transfection of siRNAs (as compared to using the nanoparticles) result in increased cell death when compared to a scrambled control sequence? Alternatively, does treatment with CA/scrambled siRNA result in no increased cell death?

Answer: Information added in line 204.

“Moreover, application of different concentrations of ‘Allstars Negative Control siRNA’ (1 pM to 10 nM) loaded into carbonate apatite structure resulted in no alteration in breast cancer cells viability.”

5)        Table 4 seems unneeded and repeats the same data as demonstrated in Figure 1 and Figure 2.

Answer: Figure 1 and 2 were eliminated from the text and data is only presented in the table format.

6)        It is not clear why 1 pM of each siRNA was used for experiments performed in Figure 3-6 when 10 or 100 pM clearly produced a more significant effect on decreased viability in Figures 1 and 2.

Answer: Information added in line 225.

“Based on extensive experiments and with the aim to explore possible synergistic effects, lowest effective doses of therapeutics were applied. This was to perhaps achieve a therapeutic effect with lower doses and hence less side effects.”

7)        Given the low dosages of chemotherapy used in Figures 3-6 (most of these compounds have reported IC50 in nM or µM range), it is not clear if the synergistic effect of co-delivery causes a significant change in drug IC50 or may be due to increased uptake of the carrier system.

Answer: In tables 5-8 the viability of the cells are compared following treatment with CA/drug vs. CA/siRNA/drug. Thus any enhancement in efficacy of the treatment could be associated with addition of the siRNA. Thereafter, the effect of the treatment at the protein level has been further explored via Western blotting. The changes in protein expression would associate the synergistic therapeutic effect with addition of siRNA.

8)        Again, Tables 5-8 seem unneeded and repeats the same data presented in Figures 3-6

Answer: Figure 3-6 were eliminated and data is only presented in the table format.

9)        In the in vivo experiments, were there any changes in body weight or were mice monitored for signs of non-specific toxicities associated with the broad distribution of particles?

Answer: Information added in line 370.

“There were no significant changes in the pattern of body weight change and also general signs and symptoms of the animals among different groups of the studies.”

10)    Were tumors histologically evaluated for overall increases in cell death or increased apoptosis?

Answer: Histological assessment of the tumors was not performed and only sizes were compared.

Minor Concerns:

1)     The manuscript should be reviewed for minor English grammatical issues to improve clarity and readability. For example in line 28, the sentence is very difficult to read.

Answer: Line 28 was amended. The manuscript was edited for grammar.

2)     Line 68 – “PI-3K” should be corrected to PI3K pathway.

Answer: Correction performed.

3)     Line 79 – chemical formula style needs to be corrected. The dot before the hydrate appears to be a subscript or a period and should be corrected as a dot. H2O should be H2O with a correct subscript. (CaCl2•2H2O)

Answer: Correction performed.

4)     Line 81 – similarly, H3PO4 needs to be formatted with the correct chemical notation including subscripts (H3PO4)

Answer: Correction performed.

5)     Line 82 – similarly, CF3COOH needs to be formatted in the correct chemical notation including subscripts (CF3COOH)

Answer: Correction performed.

6)     Line 94 – Formatting of heading should be consistent with rest of manuscript (sentence case). “Carbonate Apatite” should be lowercase.

Answer: Correction performed.

7)     Line 129 – Centrifugation speed should ideally be reported in rcf or ×g.

Answer: Description of the method was revised.

8)     Line 135 – Electrophoresis conditions are commonly reported in voltage not current. Was this experiment run under constant voltage or constant current conditions?

Answer: Description of the method was revised.

9)   Line 138 – Similar to above, electrophoresis conditions are commonly reported in voltage not current. Was the experiment run under constant voltage or constant current?

Answer: Description of the method was revised.

10)  Line 152 – Kg should be lowercase (i.e., 1.25 mg/kg of Pac)

Answer: Correction performed.

11)  Line 169 – Title in table should have “in vivo” in italics (i.e., in vivo)

Answer: Correction performed.

12)  Line 169 – Inconsistent, long spacing in line CA/AKT/ERBB2/Doc (1          mg/kg docetaxel). May be an issue with PDF generation.

Answer: Correction performed.

13)  Line 298 – Drug concentration (10 PM) should be corrected to “10 pM”

Answer: Correction performed.

14)  Line 300 – Drug concentration (10 PM) should be corrected to “10 pM”

Answer: Correction performed.

15)  Line 322 – “in vitro” and “in vivo” should be italicized

Answer: Correction performed.

16)  Figure 12 and Figure 13 should be combined as a multi-panel figure as they describe the same data.

Answer: Figure 13 was eliminated.

17)  Similarly, Figure 14 and relevant portions of Figure 16 should be combined as a multi-panel figure as they describe the same data.

Answer: Presentation of the figures was altered.

18)  Figure 15 and relevant portions of Figure 16 should be combined as a multi-panel figure as they describe the same data.

Answer: Presentation of the figures was altered.

19)  Line 401 – References to “Figure 45” and “Figure 46” should be updated appropriately.

Answer: Correction performed.

20)  Figure 17 and Figure 18 should be combined as a multi-panel figure as they describe the same data.

Answer: Presentation of the figures was altered.

21)  Line 507 – there should be consistency in the manuscript with capitalization of Western blotting.

Answer: Correction performed.

22)  Line 509 – “in vitro” and “in vivo” should be italicized.

Answer: Correction performed.

Reviewer 2 Report

This is a manuscript that describes the use of a novel and simple transfection cum drug delivery system that can deliver both siRNA and anticancer drugs together, thereby increasing the efficacy of the drugs and decreasing the toxicity associated with higher drug concentration. However the representation of the data in this manuscript is despicable. As an example, there are more than 10 figures in this manuscript, all showing MTT assay data. Its not worth showing so much of data if none of them are significant. Better representation of data is needed.

Table 4 is a little confusing. A graphical representation will be better

The representation for colors red and blue should be indicated in the figure and also in the figure legend.

None of the MTTs have significance show in the figures.

I feel like figures 1 through 9 can be condensed into a single or 2 figures to show the efficacy of siRNA treatment along with different drugs.

Each of the quadrants in figures 1 to 9 can have the drug name or siRNA name as the title to improve legibility. While the x-axis states this clearly, I encourage giving them individual titles so that the reader is not confused.

Figure 11 and 12 blots are not very convincing w.r.t knockdown. I am not convinced that the densitometry results indicated in figrue 13 are representative of the blots.

Figure 14: Why wasn't PAc alone used? The GAPDH values are also changin. This would indicate that there might not be a knockdown associated with the siRNA treatment

I am not convinced that the siRNA knockdown worked. I am also not convinced that the CA helped in the transfection.

The results seem impressive though. SO if the authors can re do the experiments to show low protein expression profile after siRNA treatment, then I am willing to go forward and accept the manuscript.

Author Response

Reviewer 2

This is a manuscript that describes the use of a novel and simple transfection cum drug delivery system that can deliver both siRNA and anticancer drugs together, thereby increasing the efficacy of the drugs and decreasing the toxicity associated with higher drug concentration. However the representation of the data in this manuscript is despicable. As an example, there are more than 10 figures in this manuscript, all showing MTT assay data. Its not worth showing so much of data if none of them are significant. Better representation of data is needed.

·         Table 4 is a little confusing. A graphical representation will be better.The representation for colors red and blue should be indicated in the figure and also in the figure legend. None of the MTTs have significance show in the figures. I feel like figures 1 through 9 can be condensed into a single or 2 figures to show the efficacy of siRNA treatment along with different drugs. Each of the quadrants in figures 1 to 9 can have the drug name or siRNA name as the title to improve legibility. While the x-axis states this clearly, I encourage giving them individual titles so that the reader is not confused.

Answer: All in vitro data were summarized for better presentation of the manuscript. The figure 1-9 are eliminated from the text and could be presented as supplementary figures.

·         Figure 11 and 12 blots are not very convincing w.r.t knockdown. I am not convinced that the densitometry results indicated in figrue 13 are representative of the blots.

Answer: Presentation of the data was revised.

·         Figure 14: Why wasn't PAc alone used? The GAPDH values are also changin. This would indicate that there might not be a knockdown associated with the siRNA treatment.

Answer: Presentation of the data was revised. Various control samples including carbonate apatite as the carrier and also CA/drug as drug loaded carrier have been included in Western blotting. This would look into any effects of each of these ingredients on the protein expression. Additionally, in another setting, samples of free drugs were used for Western blot analysis and there were no effects in the expression of the target proteins.

·         I am not convinced that the siRNA knockdown worked. I am also not convinced that the CA helped in the transfection.

Answer: The included blots demonstrate the knock down effect for various formulations of CA compared to controls. Moreover, the in vivo study further confirms the efficacy of the therapeutics.

·         The results seem impressive though. SO if the authors can re do the experiments to show low protein expression profile after siRNA treatment, then I am willing to go forward and accept the manuscript.

Answer: All experiments including protein expression and in vivo studies have been repeated several times and only reproduced results are presented here. Due to limitation with financial and human resources, repetition of the same experiments is not feasible. However, with the major corrections being performed and extensive revision of the manuscript, hopefully presentation of the data is improved.

Reviewer 3 Report

The present paper entitled « Intracellular delivery of siRNAs targeting AKT and ERBB2 genes enhances chemosensitization of breast cancer cells in culture and animal model » by Tahereh Fatemian, Hamid Reza Moghimi and Ezharul Hoque Chowdhury aimed to demonstrate that inhibition of key genes such as AKT or ERRB2 by siRNA technology potentiate the effect of chemotherapies. For that, they used carbonate apatite (CA) nanoparticles as vector and they followed cell viability in vitro and tumors growth in vivo.

Authors reported a huge amount of experiments leading to 21 figures and 7 tables. Most of them exhibited a lot of different conditions. Although the amount of work done is impressive data are not convincing. The overall effect is limited and mainly due to dose-dependent effect of chemotherapies on cell viability. Synergic effect, if any,  is really limited to very specific conditions and thus, thus should be confirm by additional experiments (n>3).  The most interesting data are in vivo effects (n=6) but those data are lost in a huge amount of in vitro experiments. In vivo effects are limited and transient (for a statistical point of view) and may be correlated with drug injections, this should be explored.

The nanoparticles are absolutely not characterized. Authors added different concentrations of siRNA and drugs in the medium before forming CA Nps but did not provide any information about NPs size, NPs number, percentage of each component loaded on NPs or remaining free. It is unclear from the manuscript if NPs are purified some way or injected as a mixture with pre-mixed compounds.

The siRNA concentration chosen for in vitro experiments (1 pM) is inefficient (as reported in Fig 2). The same team reported effects with 50 nM (Ref 12 Carbonate apatite nanoparticles carry siRNA(s) targeting growth factor receptor genes, EGFR1 and ERBB2 to regress mouse breast tumor. Tiash S., et al. 2017, Drug delivery, Vol. 24, pp. 1721-1730).   Surprisingly, it is the concentration used for in vivo experiments in Fig 19 and 20.

Figure 11, duplicate for “untreated sample” are just poor. How densitometry has been performed to obtain the corresponding value on Fig 13. Fig 13 standard errors is made on n=2 ! and statistical analysis too ! idem Figure 16 and 18: n=2.

Maybe I am wrong, maybe I understood nothing about this manuscript! but that is the point, from the current version , I am not convinced. That is the reason for not supporting it for publication in pharmaceutics.

Author Response

Reviewer 3

The present paper entitled « Intracellular delivery of siRNAs targeting AKT and ERBB2 genes enhances chemosensitization of breast cancer cells in culture and animal model » by Tahereh Fatemian, Hamid Reza Moghimi and Ezharul Hoque Chowdhury aimed to demonstrate that inhibition of key genes such as AKT or ERRB2 by siRNA technology potentiate the effect of chemotherapies. For that, they used carbonate apatite (CA) nanoparticles as vector and they followed cell viability in vitro and tumors growth in vivo.

·         Authors reported a huge amount of experiments leading to 21 figures and 7 tables. Most of them exhibited a lot of different conditions. Although the amount of work done is impressive data are not convincing. The overall effect is limited and mainly due to dose-dependent effect of chemotherapies on cell viability. Synergic effect, if any,  is really limited to very specific conditions and thus, thus should be confirm by additional experiments (n>3).  The most interesting data are in vivo effects (n=6) but those data are lost in a huge amount of in vitro experiments. In vivo effects are limited and transient (for a statistical point of view) and may be correlated with drug injections, this should be explored.

Answer:  The results of in vitro experiments were summarized and revised. All experiments including protein expression and in vivo studies have been repeated several times and only reproduced results are presented here.

·         The nanoparticles are absolutely not characterized. Authors added different concentrations of siRNA and drugs in the medium before forming CA Nps but did not provide any information about NPs size, NPs number, percentage of each component loaded on NPs or remaining free. It is unclear from the manuscript if NPs are purified some way or injected as a mixture with pre-mixed compounds.

Answer: There was a brief explanation on characterization of the nanoparticles in the original text (line 109).

“Turbidity determination and size and zeta potential measurement of the variously formulated nanoparticles have been employed for characterization of the resulting products (11).”

 Additional data on characterization of the nanoparticles was added in line 63.

“Inorganic carbonate apatite is a recently developed nanocarrier synthesized via calcium phosphate precipitation in presence of bicarbonate. The controlled crystal growth dynamic leads to formation of particles with the size ranging between 50-300 nm. These carriers encompass optimal features of efficient endocytosis, fast dissolution rate in endosomal acidic pH and effective release of loaded therapeutics. In more detail, addition of 3 mM of Ca in the synthesis of carbonate apatite results in formation of nanoparticles with the size of less than 50 nM. In presence of 10 nM paclitaxel, NPs reach the maximum size of around 170 nm. Markedly, paclitaxel loaded carbonate apatite demonstrates 20.71 ± 4.34 % loading efficiency (10). Carbonate apatite nanoparticles have been utilized for co-delivery of anti-cancer drugs and various siRNAs resulting in improved outcome (11, 12).”

Details on preparation of the injectable samples for in vivo study were revised (line 159).

The injectable nanoparticles were formulated in 100 μL of freshly prepared bicarbonated (44 mM) DMEM media to which CaCl2 was added. Samples were then incubated at 37°C for 30 min followed by maintenance on ice to prevent aggregation during injection. In drug containing samples, 1.25 mg/kg of Pac and 1 mg/kg of Doc were used prior to incubation. In case of using siRNAs, 50 nM of each siRNA was added to the media prior to incubation. The resulting therapeutics were used for iv treatment of animals.

·         The siRNA concentration chosen for in vitro experiments (1 pM) is inefficient (as reported in Fig 2). The same team reported effects with 50 nM (Ref 12 Carbonate apatite nanoparticles carry siRNA(s) targeting growth factor receptor genes, EGFR1 and ERBB2 to regress mouse breast tumor. Tiash S., et al. 2017, Drug delivery, Vol. 24, pp. 1721-1730).   Surprisingly, it is the concentration used for in vivo experiments in Fig 19 and 20.

Answer: Additional explanation added in line 225.

“Based on extensive experiments and with the aim to explore possible synergistic effects, lowest effective doses of therapeutics were applied. This was to perhaps achieve a therapeutic effect with lower doses and hence less side effects.”

·         Figure 11, duplicate for “untreated sample” are just poor. How densitometry has been performed to obtain the corresponding value on Fig 13. Fig 13 standard errors is made on n=2 ! and statistical analysis too ! idem Figure 16 and 18: n=2.

Answer:  Presentation of the data was revised.

·         Maybe I am wrong, maybe I understood nothing about this manuscript! but that is the point, from the current version , I am not convinced. That is the reason for not supporting it for publication in pharmaceutics.

Answer: With the major corrections being performed and extensive revision of the manuscript, hopefully presentation of the data is improved.

Round 2

Reviewer 1 Report

The quality of the manuscript has been significantly improved through the provided revisions. I find no other concerns that were not already addressed previously.

Author Response

Thank you.

Reviewer 2 Report

Accept

Author Response

Thank you.

Reviewer 3 Report

The manuscript « Intracellular delivery of siRNAs targeting AKT and ERBB2 genes enhances chemosensitization of breast cancer cells in culture and animal model » by Tahereh Fatemian, Hamid Reza Moghimi and Ezharul Hoque Chowdhury has been considerably improved  mainly by removing most of the figures of the first part and by providing a synthetic table of the data and a consistent discussion. The Material and Method section has been also improved. The third part reporting in vivo experiments remained interesting.

However, western blot-based data remained problematic as they did not fit with the usual requirements for this kind of results. I did not find specific requirements for western blot experiments in “   Pharmaceutics  “ but I referred to:

(1)Brooks HL & Lindsey ML (2018). Guidelines for authors and reviewers on antibody use in physiology studies. Am J Physiol Heart Circ Physiol 314, H724–H732.

(2) Nature research group editorial policies

https://www.nature.com/nature-research/editorial-policies/image-integrity

In the present version of the manuscript loading marker has been removed (?) and the sample remained in duplicate (3 values are mandatory for any quantitative analysis). With only 2 samples, how means & standard deviations has been calculated? How statistical analysis has been performed with 2 values?  The former fig 13, 16 and 18 have been removed but the data and the statistical analyses remain in the text and do support the author’s conclusions.

In some journal, a complete picture of the gel/dot should be provided in supplementary data, in some other theses pictures should be provided to the reviewers, or editor. These pictures should include in the same gel/dot mass markers, loading markers .., etc. (see guidelines).

For these reason, I don’t recommend the present paper for publication in “Pharmaceutics”

Author Response

Reviewer 3: However, western blot-based data remained problematic as they did not fit with the usual requirements for this kind of results. I did not find specific requirements for western blot experiments in “   Pharmaceutics  “ but I referred to:

(1)Brooks HL & Lindsey ML (2018). Guidelines for authors and reviewers on antibody use in physiology studies. Am J Physiol Heart Circ Physiol 314, H724–H732.

(2) Nature research group editorial policies

https://www.nature.com/nature-research/editorial-policies/image-integrity

In the present version of the manuscript loading marker has been removed (?) and the sample remained in duplicate (3 values are mandatory for any quantitative analysis). With only 2 samples, how means & standard deviations has been calculated? How statistical analysis has been performed with 2 values?  The former fig 13, 16 and 18 have been removed but the data and the statistical analyses remain in the text and do support the author’s conclusions.

In some journal, a complete picture of the gel/dot should be provided in supplementary data, in some other theses pictures should be provided to the reviewers, or editor. These pictures should include in the same gel/dot mass markers, loading markers .., etc. (see guidelines).

For these reason, I don’t recommend the present paper for publication in “Pharmaceutics”

Answer:

The images for loading marker bands were removed based on the findings on significant variation in the expression level of some housekeeping genes following different conditions of the cells including cancer (reference to A&B).

A. An old method facing a new challenge: re-visiting housekeeping proteins as internal reference control for neuroscience research. Rena Li and Yong Shen. 2013, Life Sci., Vol. 92(13), pp. 747–751.

B. Common housekeeping proteins are upregulated in colorectal adenocarcinoma and hepatocellular carcinoma, making the total protein a better "housekeeper". Xiaowen Hu., et al. 2016, Oncotarget, Vol 7(41), pp. 66679–66688.

Reference to Nature research group editorial policies for image integrity and standards (C), statistical analysis for Western blot data was performed using results from triplicate samples derive from different blots of the same experiment processed in parallel. Thus, p values in line 313 and 323 were revised to < 0.05.

C. https://www.nature.com/nature-research/editorial-policies/image-integrity

The images included for Western blot results all belong to a single blot with no vertical slicing. Additionally, all images for loading controls are run on the same blot and only cropped to be exposed to different antibodies. However, for more complete images, I would require a longer time to have access to my old archive on a pc in another country.

Round 3

Reviewer 3 Report

Housekeeping protein is not perfect but no protein at all is worse.
Alternative methods exist such as staining all proteins into the gel !
n=2 remains n=2 and statistical analysis could not be performed.
Conclusions are not supported.